

# Estimation of isotopologue variation of N₂O during denitrification by *Pseudomonas aureofaciens* and *Pseudomonas chlororaphis*: Implications for N₂O source apportionment

Joshua A Haslun[1, 3], Nathaniel E. Ostrom[2, 3], Eric L. Hegg[1, 3], Peggy H. Ostrom[2, 3]

[1]Biochemistry and Molecular Biology, Michigan State University, East Lansing, 48824, USA
[2]Integrative Biology, Michigan State University, East Lansing, 48824, USA
[3]Great Lakes Bioenergy Research Center, Michigan State University, East Lansing, 48824, USA

*Correspondence to*: Peggy H. Ostrom, Nathaniel E. Ostrom

**Abstract.** Soil microbial processes, stimulated by agricultural fertilization, account for 90 % of anthropogenic nitrous oxide ($N_2O$), the leading source of ozone depletion and a potent greenhouse gas. Efforts to reduce $N_2O$ flux commonly focus on reducing fertilization rates. Management of microbial processes responsible for $N_2O$ production may also be used to reduce $N_2O$ emissions, but this requires knowledge of the prevailing process. To this end, stable isotopes of $N_2O$ have been applied to differentiate $N_2O$ produced by nitrification and denitrification. To better understand the factors contributing to isotopic variation during denitrification, we characterized the $\delta^{15}N$, $\delta^{18}O$ and site preference (SP; the intramolecular distribution of $^{15}N$ in $N_2O$) of $N_2O$ produced during $NO_3^-$ reduction by *Pseudomonas chlororaphis subsp. aureofaciens* and *P. c. subsp. chlororaphis*. In addition to species, treatments included electron donor (citrate and succinate) and electron donor concentration (0.01 mM, 0.1 mM, 1 mM, and 10 mM) as factors. In contrast to the expectation of a Rayleigh model, all treatments exhibited curvilinear behaviour between $\delta^{15}N$ or $\delta^{18}O$ and [-flnf/(1-f)]. The curvilinear behaviour indicates that the fractionation factor changed over the course of the reaction, something that is not unexpected for a multi-step process such as denitrification. Using the derivative of the equation, we estimated that the net isotope effects (η) vary by as much as 100 ‰ over the course of a single reaction, placing challenges for using $\delta^{15}N$ and $\delta^{18}O$ as apportionment tools. In contrast, SP for denitrification was not affected by the extent of the reaction, the electron donor source, or concentration, although the mean SP of $N_2O$ produced by each species differed. Therefore, SP remains a robust indicator of the origin of $N_2O$. To improve apportionment estimates with SP, future studies could evaluate other factors that contribute to the variation in SP.

## 1 Introduction

Agricultural production of food and energy has required a 10-fold increase (i.e. from 10 to 100 TGN yr⁻¹) in the application of synthetic fertilizer since 1950 (Robertson and Vitousek, 2009). Moreover, to maximize crop yields, nitrogen (N) is often applied at rates in excess of a crop's yield response, the average maximum crop yield as a function of fertilizer application rate. This results in a residual N pool (Sebilo et al., 2013). While some of the excess N may be incorporated into the soil, much





of it is either transported out of the system via runoff as $NO_2^-$ or $NO_3^-$ (Zhou and Butterbach-Bahl, 2014), volatilized as $NH_3$ (Pan et al., 2016), or converted to $N_2$ and/or nitrous oxide ($N_2O$) via oxidative and reductive microbial processes (Schreiber et al., 2012; Venterea et al., 2012) such as nitrification, and denitrification, respectively. Stimulated by agricultural practices, these microbial processes account for 90 % of anthropogenic $N_2O$ (Denman, 2007; Reay et al., 2012). Losses of N from soils

in the form of $N_2O$ are of particular concern because this greenhouse gas contributes to stratospheric ozone depletion (Portmann et al., 2012; Ramanathan et al., 1985; Ravishankara et al., 2009) and has a 100-year global warming potential that is approximately 300 times that of $CO_2$ (IPCC, 2014). Moreover, the relationship between N application rate and $N_2O$ emissions from agricultural soils is non-linear (McSwiney and Robertson, 2005; Shcherbak et al., 2014), with $N_2O$ emissions dramatically increasing with moderate increases in fertilization. To mitigate $N_2O$ flux without compromising crop yield, several systems

have been developed that includes, the maximum return to nitrogen system (Nafziger et al., 2004) and the variable rate nitrogen application system (Scharf et al., 2011). These strategies provide recommendations of fertilization rates that minimize reductions in crop yield while simultaneously decreasing the amount of residual N available for $N_2O$ production, thereby lowering soil $N_2O$ flux. Identifying how to manage the microbial processes contributing to $N_2O$ flux from agricultural systems would be an additional mechanism to mitigate atmospheric $N_2O$ additions (Paustian et al., 2016; Reay et al., 2012; Venterea

et al., 2012). Because nitrification and denitrification require aerobic and anaerobic conditions, respectively, strategies directed at controlling soil oxygen saturation could become part of a management strategy (Kravchenko et al., 2017). This, however, requires identifying the relative importance of nitrification and denitrification to $N_2O$ flux spatially and temporally across different agricultural landscapes.

The stable isotope ratios of $\delta^{15}N$ and $\delta^{18}O$ have been used to apportion $N_2O$ flux between nitrification and denitrification

(Davidson and Keller, 2000; Park et al., 2011; Yamagishi et al., 2007). Apportionment approaches require that the isotope values of $N_2O$ differ between the two production processes and remain constant throughout the course of a reaction (Jinuntuya-Nortman et al., 2008; Ostrom and Ostrom, 2017). However, shifts or fractionation in the isotope values of $N_2O$ produced during either nitrification or denitrification can compromise source apportionment (Barford et al., 1999; Perez et al., 2000; Sutka et al., 2008; Yoshida, 1988), and reduction of $N_2O$ by denitrification may further alter isotope values (Jinuntuya-Nortman

et al., 2008). Nevertheless, $\delta^{15}N$ and $\delta^{18}O$ can still be a useful tool if environmental conditions constrain processes, such as anoxic conditions prohibiting nitrification.

Site preference (SP), the difference in $^{15}N$ abundance between the central N ($\delta^{15}N^\alpha$) and terminal N ($\delta^{15}N^\beta$) of $N_2O$, offers an alternative tool for apportionment of $N_2O$ production (Yoshida and Toyoda, 2000). The large difference in SP of $N_2O$ produced from nitrification and denitrification (ca. 30 ‰) paired with the observations that SP is constant during $N_2O$ production, and

is independent of the isotopic composition of the nitrogen substrates of nitrification and denitrification, has prompted the use of SP for $N_2O$ source apportionment (Sutka et al., 2006; Toyoda et al., 2005). However, SP is not without variation (Toyoda et al., 2017). In addition to production pathway, numerous factors could theoretically control the degree of variation in SP including differences in bacterial species, the specific enzyme involved in its production (Yang et al., 2014), and, for





denitrification, carbon source. The accuracy of apportionment estimates using isotope values, including SP, will be improved by understanding sources of variation.

This study investigated the effect of carbon source (electron donor), and carbon source concentration on $\delta^{15}N$ and $\delta^{18}O$ of $N_2O$ produced by two denitrifier species in vitro, as well the effect of these factors on SP values of $N_2O$. We conducted our study

with *Pseudomonas chlororaphis subsp. chlororaphis* and *P. chlororaphis subsp. aureofaciens* because they are highly related denitrifiers that lack $N_2O$ reductase, but encode different nitrite reductases (NIR).

## 2 Materials and Methods

### 2.1 Organisms and Culture Conditions

Cultures of *Pseudomonas chlororaphis subsp. chlororaphis* (ATCC 43928; *P. chlororaphis*) and *Pseudomonas chlororaphis*

*subsp. aureofaciens* (ATCC 13985; *P. aureofaciens*) were cryogenically stored (-80 °C) in tryptic soy broth (TSB; Caisson Labs, Smithfield, UT) and sterile glycerol 1:1 (v/v). Stock cultures were re-established in 5 mL TSB amended with sodium nitrate (NaNO$_3$, 10 mM; Sigma-Aldrich, St. Louis, MO) under aerobic conditions at a constant temperature with continuous agitation (18 h, 25 °C). Individual colonies were obtained from re-established stock cultures by the streak-plate technique on tryptic soy agar (TSA; Caisson Labs, Smithfield, UT) amended with NaNO$_3$ (10 mM). Tryptic soy agar plates of stock cultures

were sealed with parafilm and incubated (aerobic, 25 °C). The plates were stored at 4 °C for up to two weeks prior to establishment in liquid media for denitrification experiments.

### 2.2 Preparation of Cultures for Denitrification Experiments

Starter cultures of each species were established in 5 mL TSB amended with NaNO$_3$ (10 mM) with 1 colony from stored stock culture plates (Thermo Fisher Scientific, Waltham, MA). Cultures were then grown aerobically with agitation (25 °C, 18 h) to

late exponential phase (optical density at 600 nm (OD$_{600}$) = 0.3). Optical density was determined with a Spectronic 20 spectrophotometer (Bausch and Lomb, Rochester, NY). Two 160 mL sterile serum bottles containing 50 mL of carbon minimal media (CMM) (Anderson et al., 1993) amended with 10 mM NaNO$_3$ and 10 mM sodium succinate (Sigma-Aldrich, St. Louis, MO) were each inoculated with 200 µL of the aerobic culture. The bottles were stoppered (Geomicrobial Technologies, Inc.), crimp sealed, and the headspace sparged with ultra-high purity (UHP) N$_2$ for 15 min. Cultures were incubated (25 °C, 18 h)

with agitation. Following 18 h, the cells were transferred to 50 mL conical Falcon™ tubes (Corning, Corning, NY) and centrifuged (3,000 x *g*, 30 min, 25 °C) to pellet the cells. The supernatant was decanted and the cells dispersed in CMM lacking a carbon or nitrogen source (OD$_{600}$ = 0.2). The cells were aliquoted (2 mL) into sterile 35 mL serum bottles, which were then stoppered (Geomicrobial Technologies, Inc.) and crimp sealed. An anaerobic environment was created by sparging the cells with UHP N$_2$ for 20 min. Sparging was accomplished by inserting one sterile stainless-steel needle (#20 Thomas Scientific,

Swedesboro, NJ) carrying N$_2$ through the stopper into the media while a second sterile stainless-steel needle was inserted through the stopper and into the headspace to allow gas to exit. Following sparging, the bottles were allowed to reach





atmospheric pressure, and reactions were then initiated by injecting 20 μL of the carbon source (anaerobic) to reach a final concentration of 0.01 mM, 0.1 mM, 1 mM, or 10 mM. Treatments with citrate and succinate concentrations of 1 mM and 10 mM were conducted for both bacterial taxa. Treatments of citrate and succinate at 0.1 mM were only conducted for *P. chlororaphis*. Treatments with a carbon source concentration of 0.01 mM were only conducted with succinate but were done

so for both taxa. The addition of the carbon source was followed by adding 26 μL of 0.1 M NaNO$_3$ (anaerobic) to reach a final NO$_3^-$ concentration of 1.3 mM.

**2.3 Isotope Analysis and Modelling Isotope Behaviour**

Each treatment consisted of four denitrification cultures. Headspace samples were obtained from each culture with a gas tight syringe (Hamilton; Reno, NV). For one of the four cultures, a 100 μL headspace sample was obtained every 15 minutes for

analysis of N$_2$O concentration. Headspace N$_2$O concentration of this culture was determined with a Shimadzu Greenhouse Gas Analyzer gas chromatograph equipped with an electron capture detector (ECD) (model GC-2014, Shimadzu Scientific Instruments; Columbio, MD). For details regarding this method see Yang et al. (2014). These data were used to determine when the N$_2$O concentration was sufficient for isotope analysis and to estimate the volume of headspace required for isotope analysis over the course of the reaction. Headspace sampling of the remaining three cultures was initiated when the N$_2$O

concentration determined by ECD was above ca. 0.4 ppm. Headspace samples of each of the 3 cultures were injected into 60 ml serum bottles (one per culture) that had been sparged with UHP N$_2$ for 15 min, and stored for isotope analysis. Samples were analyzed on an IsoPrime100 stable isotope ratio mass spectrometer (IRMS) interfaced to a TraceGas inlet system (Elementar; Mt. Laurel, NJ) (Sutka et al., 2003). The inlet system used He as the carrier gas and removed both water and CO$_2$ with separate magnesium perchlorate (Costech; Valencia, CA) and CO$_2$ absorbent traps (Carbosorb, 8-14 mesh, Costech;

Valencia, CA), respectively, prior to concentrating N$_2$O within a cryofocusing trap. Chromatographic separation of N$_2$O was achieved with a Porplot Q column prior to isotopic analysis. Mass overlap and related corrections followed the protocol outlined in Toyoda and Yoshida (2000). The mean precision of replicate N$_2$O standards were 0.1 ± 0.1 ‰, 0.3 ± 0.18 ‰, 0.3 ± 0.17 ‰, 0.2 ± 0.1 ‰, and 0.6 ± 0.3 ‰ composition for $\delta^{15}N$, $\delta^{15}N^\alpha$, $\delta^{15}N^\beta$, $\delta^{18}O$, and SP, respectively.

The $\delta^{15}N$ and $\delta^{18}O$ values are reported as:

$$\delta = \left[\left(\frac{R_{sample}}{R_{standard}}\right) - 1\right] \times 1,000, \tag{1}$$

where R is the ratio of the trace to the abundant isotope of N or O, and air and VSMOW are the standards for N and O, respectively. Site preference is defined as

$$SP = \delta^{15}N^\alpha - \delta^{15}N^\beta, \tag{2}$$

where $\delta^{15}N^\alpha$ and $\delta^{15}N^\beta$ are the isotope values at the central and peripheral N atom of the linear N$_2$O molecule, respectively.

The changes in $\delta^{15}N$, $\delta^{18}O$, and SP of N$_2$O during the course of the reaction were investigated using the Rayleigh equation by plotting each isotope value vs. [-flnf/(1-f)] where f is the fraction of substrate remaining (Mariotti et al., 1981). Generalized additive modelling of the relationship between the isotope value of N$_2$O and [-flnf/(1-f)] indicated asymptotic curvilinear



behaviour. Therefore, we performed non-linear least squares regression starting with a three-parameter exponential function of the form

$$y = a + ce^{b[x]}. \tag{3}$$

Model reduction and selection were performed following the methods of Baty et al., (2015). Non-linear model fit was also

compared to a linear model fit. Models with the lowest residual standard error, fewest iterations to convergence (< 10), lowest parameter confidence intervals, and lowest collinearity of variables were deemed to have the best fit. The goodness of fit for each model was also assessed visually from residual plots. Model residuals that displayed patterns were also deemed poor. This process produced an exponential function with the generalized form

$$y = a + e^{b[x]}, \tag{4}$$

where y is the isotope value of the accumulated product, x is [-flnf/(1-f)], and a and b are coefficients estimated by the model. The starting values supplied to the function were a = 7 and b = 5 for $\delta^{15}N$ and a = 75 and b =10 for $\delta^{18}O$. These starting values were selected because they are greater than the expected coefficients, which aids in model convergence (Baty et al., 2015). The derivative of Eq. 4,

$$y' = be^{b[x]} \tag{5}$$

can be used to predict the slope at any extent of the reaction. The term net isotope effect ($\eta$) has been used to describe isotopic discrimination, the change in isotope value, observed during a multi-step reaction (Jinuntuya-Nortman et al., 2008). Therefore, $\eta$ is equivalent to y' in Eq. 5.

We used probability density functions to illustrate the probability density distribution (PDD) of $\eta$ across the extent of the reaction observed. Probability density functions were determined with a Gaussian smoothing kernel from 50 equally spaced

estimates of $\eta$ spanning the complete extent of the reaction (i.e. f = 0 to 1). The bandwidth was set to 1 for each density estimate.

Modelling was performed with R statistical software (Team, 2013), and all figures were produced with ggplot2 (Wickham, 2009, 2011).

## 2.4 Statistical Analysis of SP Data

We used a linear model to determine if SP changed as a function of [-flnf/(1-f)]. Significant relationships were not observed and therefore the effect of taxa, carbon source, and carbon source concentration on mean SP was examined with Analysis of Variance (ANOVA). Tukey's HSD was used to identify significant differences between and among groups. Normality of the data was assessed with Q-Q plots and the Shapiro-Wilk test.

Statistical analyses were performed with R statistical software (Team, 2013), and all figures were produced with ggplot2 within

that software platform (Wickham, 2009, 2011).



## 3 Results

### 3.1 Effect of Carbon Source and Concentration on $\delta^{15}$N-N$_2$O

The $\delta^{15}$N of N$_2$O produced by the two denitrifier taxon in our study produced a non-linear relationship with the fraction of substrate remaining expressed in the Rayleigh model as [-flnf/(1-f)] (Figure 1). The derivative, Eq. (5), of the exponential

equation for the curvilinear relationship between isotope value and [-flnf/(1-f)] Eq. (4), indicated that $\eta^{15}$N changed over the course of the reaction (Figure 1, Supplementary Table 1). Estimates of $\eta^{15}$N during denitrification of NO$_3^-$ to N$_2$O by *P. aureofaciens* ranged from -77.5 ‰ to -18.4 ‰ and -106.2 ‰ to -11.4 ‰ for citrate and succinate, respectively, while values ranged from -119 ‰ to -9.2 ‰ and -82.1 ‰ to -5.1 ‰ for citrate and succinate, respectively, during denitrification by *P. chlororaphis* (Figure 2, Supplementary Table 1). Probability density distributions of $\eta^{15}$N for all treatments show that the

majority of values are of lower magnitude, and values between -50 ‰ and -10 ‰ were most probable. High magnitude values for $\eta$ occurred at the beginning of reactions, where values of [-flnf/(1-f)] are high (i.e. closer to 1).

### 3.2 Effect of Carbon Source and Concentration on $\delta^{18}$O-N$_2$O

The $\delta^{18}$O of N$_2$O produced by the two taxa displayed a non-linear relationship with [-flnf/(1-f)] (Figure 1), and much like $\delta^{15}$N, an exponential model Eq. (4) was the most parsimonious fit. The exponential equations determined for each treatment along

with the derivatives are presented in Supplementary Documents (Table 2). Similar to the variation in $\eta^{15}$N, the most rapid changes in $\eta^{18}$O occurred early in the extent of the reactions (i.e. larger values of [-flnf/(1-f)]). Estimates of $\eta^{18}$O determined following denitrification of NO$_3^-$ by *P. aureofaciens* ranged from -22.2 ‰ to -9.8 ‰ and -77.0 ‰ to -3.1 ‰ for citrate and succinate, respectively, while the reduction of NO$_3^-$ to N$_2$O by *P. chlororaphis* produced $\eta^{18}$O values that ranged from -75.4 ‰ to -7.5 ‰ and -67.8 ‰ to -4.0 ‰ for citrate and succinate, respectively. Probability density distributions of $\eta^{18}$O

indicated that treatments with narrow observed f ranges were not strictly associated with narrow PDDs (Figures 1 and 3). For instance, the observed f range for *P. aureofaciens* reduction with 0.01 mM succinate was nearly 0.6 while the range in $\eta^{18}$O was less than -10 ‰. The rate of change of the function's slope is controlled by parameter b in Eq. (4). Therefore, lower estimates of parameter b produce narrow ranges of $\eta^{18}$O, such as that observed for *P. aureofaciens* reactions with a succinate concentration of 0.01 mM.

### 3.3 Site Preference as a Function of Carbon-Source and Concentration

Site preference did not change as a function of the extent of the reaction, and across all treatments SP ranged from -7.0 ‰ to 6.0 ‰ (Figure 1). Denitrification by *P. aureofaciens* produced a mean SP of 0 ‰ (st. dev. = 3.3 ‰); however negative SPs observed at 1 mM succinate (mean = -4.2 ‰, st. dev. = 1.8 ‰) contributed greatly to this value (Figure 4). Denitrification of NO$_3^-$ by *P. chlororaphis* produced mean SP values that were similar among all carbon source treatments; -3.7 ‰ (st. dev. =

2.2 ‰) and -4.2 ‰ (st. dev. = 1.2 ‰) for citrate and succinate, respectively.

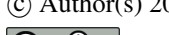



Analysis of variance identified a significant difference in SP values among the treatments examined for each species (ANOVA, p < 0.001). Across all treatments, the average SP of *P. chlororaphis* denitrification was 4.1 ‰ lower than that of *P. aureofaciens*. In addition to taxon, the carbon source (i.e. succinate or citrate) also contributed somewhat to differences in SP between treatments (ANOVA; p < 0.01) with growth on succinate producing SP values 0.9 ‰ lower than those produced with

citrate as the carbon source. Interestingly, the concentration of the carbon source had no discernible effect on SP under our reaction conditions. Therefore, the variation in SP was largely dependent on taxa (Figure 4).

## 4 Discussion

This study investigated the effect of denitrifier species, carbon source (electron donor), and electron donor concentration on $\delta^{15}N$, $\delta^{18}O$, and SP isotope values of $N_2O$ produced during denitrification in pure culture. We observed isotopic discrimination

against $^{15}N$ and $^{18}O$ but no change in SP during the reduction of $NO_3^-$ to $N_2O$ by *P. aureofaciens* or *P. chlororaphis,* and these observations held regardless of carbon source and electron donor concentration.

We modelled isotope discrimination within each of our experiments. According to convention (Mariotti et al., 1981), the magnitude of the isotopic fractionation factor ($\alpha$) for a single unidirectional reaction is defined by the rate constants of the light ($k_1$) and heavy ($k_2$) isotopically substituted compounds:

$$\alpha = k_2/k_1. \tag{6}$$

Further, the isotopic enrichment factor, $\varepsilon$, is defined as

$$\varepsilon = (\alpha - 1) \times 1000, \tag{7}$$

and can be estimated from the slope of the linear relationship described by the Rayleigh model:

$$\delta^{15}N_p = \delta^{15}N_{so} - \varepsilon_{\frac{p}{s}}[(flnf)/(1-f)]; \tag{8}$$

where $\delta^{15}N_p$ is the isotope value of the accumulated product, $\delta^{15}N_{so}$ is the isotope value of the initial substrate, $\varepsilon$ is the fractionation factor, and f is the fraction of substrate remaining (Mariotti et al., 1981). However, the reduction of $NO_3^-$ to $N_2O$ by *P. aureofaciens* and *P. chlororaphis*, displayed a non-linear exponential relationship between $\delta^{15}N$ vs. [-flnf/(1-f)] and $\delta^{18}O$ vs. [-flnf/(1-f)]. This curvilinear isotopic behaviour was evident for denitrification metabolizing both carbon substrates (citrate or succinate) and at all substrate concentrations (Figure 2, Supplementary Table 1). The non-linear behaviour indicates that

the fractionation factor, $\varepsilon$, is not constant, a phenomenon not unexpected for multi-step reactions in which more than one enzymatic step and diffusion of products and/or substrates into and out of the cell can result in variation in isotopic discrimination (Granger et al., 2008; Sutka et al., 2008). Because the fractionation factor varies during multi-step reactions, it is best considered a net isotope effect ($\eta$) (Jinuntuya-Nortman et al., 2008). The reduction of $NO_3^-$ to $N_2O$ during denitrification involves three enzymes and multiple opportunities for diffusion, all cases where isotope discrimination can occur (Figure 5).

Similar to other studies, our previous work on denitrification estimated $\eta$ from a Rayleigh model (Barford et al., 1999; Lewicka-Szczebak et al., 2014; Sutka et al., 2006; Toyoda et al., 2005; Yano et al., 2014). However, the Rayleigh model




assumes a unidirectional single-step reaction with linear behaviour, assumptions that are clearly not valid for $N_2O$ production from nitrate during denitrification. Thus, here we developed estimates of η from the derivative of the exponential relationship between the isotope value of the accumulated product, $N_2O$, and the extent of the reaction [-f lnf/(1-f)]. This allowed us to quantify changes in η over the course of the denitrification reaction.

For our entire data set, $\eta^{15}N$ and $\eta^{18}O$ varied by as much as ca. 100 ‰ within a single experiment (Figures 2, 3). Note, however, that during $N_2O$ production, both $\delta^{18}O$ and $\eta^{18}O$ can be influenced by oxygen exchange between water and nitrogen oxides (Kool et al., 2009, 2011). These oxygen exchange effects are difficult to quantify making interpretation of $\eta^{18}O$ data difficult. Thus, we limit our discussion of fractionation to $\eta^{15}N$. Values of $\eta^{15}N$ previously reported for reduction of $NO_3^-$ to $N_2O$ in pure cultures (-43 ‰ to -9 ‰) fall within the range we observed (Sutka et al., 2006; Toyoda et al., 2005; Rohe et al. 2014; Sutka et al 2008). However, some of our values are much greater in magnitude than those previously reported (e.g. -119 ‰). Values of
such magnitude occurred near the onset of the reaction (i.e. high values of [-flnf/(1-f)]), most notably when no more than 10 % of the $NO_3^-$ had been reduced. The occurrence of high magnitude η values near the beginning of the reaction is likely related to the relative importance of diffusion and enzymatic fractionation in controlling η. Fractionation associated with enzymes is often much larger than that associated with diffusion, and enzymatic fractionation is fully expressed when diffusion does not
limit substrate supply to the enzyme (Jinuntuya-Nortman et al., 2008; Ostrom and Ostrom, 2012). Thus, the largest η is expected at the beginning of the reaction, consistent with what we observed. Large magnitude values for η can be easily missed if the isotope value of the accumulated product is used to estimate η. Without knowledge of production rate, it can be difficult to know when there is sufficient product for isotopic measurement. By characterizing production rates before initiating experiments to estimate η, we were able to capture isotope values for $N_2O$ close to the onset of the reaction.

There are important reasons why published discrimination factors might be less negative and therefore of lower magnitude than ours. Prior estimates were derived from a single slope from a Rayleigh model and, therefore, do not produce estimates of η over the course of the reaction. Importantly, they may not characterize the large fractionation occurring at the onset of a reaction. Even so, our highly negative values for η might, initially, seem remarkable. Considering variation in η in the context of a multi-step model provides insight into how these values might arise, particularly in the early stages of a culture when the
substrate concentration is high. The reduction of $NO_3^-$ to $N_2O$ includes three enzymatic steps in which substantive fractionation may occur (Figure 5). As a consequence, we would expect the products of each successive reaction to become progressively depleted in the heavy isotope, assuming normal ε. If, for example, the ε for each of the three steps was -40 ‰ then reduction of nitrate with a $\delta^{15}N$ of 0 ‰ could yield $N_2O$ of -120 ‰. Thus, denitrification has the potential to produce $N_2O$ that is greatly depleted in $^{15}N$ resulting in highly negative values for η. As the reaction proceeds, each enzyme is likely to be limited by the
supply of substrate from diffusion. This has a tendency to reduce expression of fractionation, and η is therefore reduced to less negative values.

Probability density distributions indicate that markedly low η values associated with one endpoint of the range in η are not common (Figure 3). They also illustrate the range in η that would be expected for the reaction, and their shape emphasizes



important changes in η during the course of a reaction. For example, several of the distributions show a marked change in slope on the left side of the distribution (e.g. 10 mM citrate $\eta^{15}N$, both species) that is a consequence of a significant change in slope along the curve of $\delta^{15}N$ vs. [-flnf/(1-f)] (Figures 1, 3). While we cannot ascribe a specific event to this change, future studies aimed at investigating specific enzymes may provide a better understanding of the behaviour of η during denitrification.

Perhaps most importantly, these distributions emphasize that assessments of net isotope effects for multi-step reactions will not be complete without consideration of isotopic behaviour over a wide extent of the reaction and the development of models that describe isotope behaviour that does not fit a linear Rayleigh model.

In contrast to the results we observed for $\delta^{15}N$ and $\delta^{18}O$, isotopic discrimination was not evident for SP regardless of treatment (Figure 2). Instead, SP was constant during the course of the reaction. This finding is consistent with pure culture studies of

nitrification and denitrification across multiple species (Frame and Casciotti, 2010; Sutka et al., 2003, 2006; Toyoda et al., 2005). The differences we observed in SP between species, however, relates to the factors that control SP. Unlike the case for bulk isotopes, SP is determined during a single reaction, the reduction of NO to $N_2O$ (Toyoda et al., 2005). Thus, SP is only influenced by nitric oxide reductase (NOR) activity and diffusion of NO or $N_2O$ into or out of the cell. If we solely consider enzymatic fractionation by NOR, SP is related to three factors; the fraction of substrate remaining f, $\delta^{15}N^{\alpha}$, and $\delta^{15}N^{\beta}$. In this

case, however, f differs from the quantity we refer to in Fig. 1, which is the fraction of $N_2O$ produced in a multi-step reaction. For the purposes of discussing solely NO reduction, we are unable to use the concentration of the end product of a multi-step reaction. Instead, we could define f as either the concentration of substrate (i.e. NO) available to NOR or the concentration of product (i.e. $N_2O$) produced from the reaction mediated by NOR. While we did not measure the concentration of the substrates or products of NO reduction in bacterial cells directly, to understand isotope systematics we can consider their behaviour in

the environment. Our considerations will focus on NO. The NO concentration will depend on its production, reduction, and losses due to diffusion into or out of the cell, all of which could vary between species. We do not know, for example, the degree to which the rate of NO production intrinsically differs between the $cd_1$-type NIR of *P. chlororaphis* and copper containing NIR of *P. aureofaciens* or how gene expression may alter these rates. We posit that small differences in SP between and even within species in our study and others may relate to the size of the NO pool available to NOR.

Nitrous oxide is the third most abundant greenhouse gas in the atmosphere and is the greatest source of stratospheric ozone depletion (Ravishankara et al., 2009). Moreover, efforts to balance the $N_2O$ budget have been challenged by the episodic nature of $N_2O$ flux (Nishimura et al., 2005) and, historically, identifying the pathway of $N_2O$ production has been enigmatic (Schreiber et al., 2012). Here we emphasize that within our toolbox, SP remains a robust indicator of $N_2O$ derived from denitrification regardless of carbon source or concentration, and we identify that a component of the variation in SP can be ascribed to species

differences. Our ability to understand factors that control variation in SP is important to refining estimates of the relative importance of $N_2O$ production pathways, something that is necessary for mitigation of fluxes of this important GHG from aquatic and terrestrial environments.





**Acknowledgments**

This work was funded by the DOE Great Lakes Bioenergy Research Center (DOE BER Office of Science DE-FC02-07ER64494). We would like to thank Dr. Hasand Gandhi from Michigan State University for his insight and direction during isotope analyses.

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



**Figure 1:** δ$^{15}$N, δ$^{18}$O, and site preference (SP) of N$_2$O produced during denitrification of NO$_3^-$ by *Pseudomonas chlororaphis subsp. aureofaciens* and *Pseudomonas chlororaphis subsp. chlororaphis* with different electron donor sources and concentrations. A larger value of [-flnf/(1-f)], where f is the fraction of substrate remaining, represents earlier points in the reaction. The curved relationships are of the form $y = a + e^{b[x]}$, where y is the isotope value, x is [-flnf/(1-f)] and a and b are the estimated coefficients that affect the y-intercept and curvilinear shape, respectively.





**Figure 2. Probability density distributions (PDD) of δ$^{15}$N net isotope effects (η) derived from the derivative of the exponential function (Eq. 4) describing the relationship between δ$^{15}$N and [-flnf/(1-f)] for *Pseudomonas aureofaciens* (orange) and *Pseudomonas chlororaphis* (blue). Estimates of η were produced over the entire extent of the reaction (i.e. f=0 to 1). The left panel displays the**
5 **PDDs for citrate treatments and the right for succinate treatments. Positive values of η were not observed during reactions. Tally marks at the base of each panel indicate the actual distribution of calculated values.**



**Figure 3. Probability density distributions (PDD) of δ[18]O net isotope effects (η) estimated from the derivative of an exponential function describing the relationship between δ[18]O and [-flnf/(1-f)] for *Pseudomonas aureofaciens* (orange) and *Pseudomonas chlororaphis* (blue). The estimates of η are extrapolated to include the complete extent of the reaction. The left panel displays the PDDs for citrate treatments and the right for succinate treatments. Positive values of η were not observed during reactions. Tally marks at the base of each panel indicate the actual distribution of calculated values.**





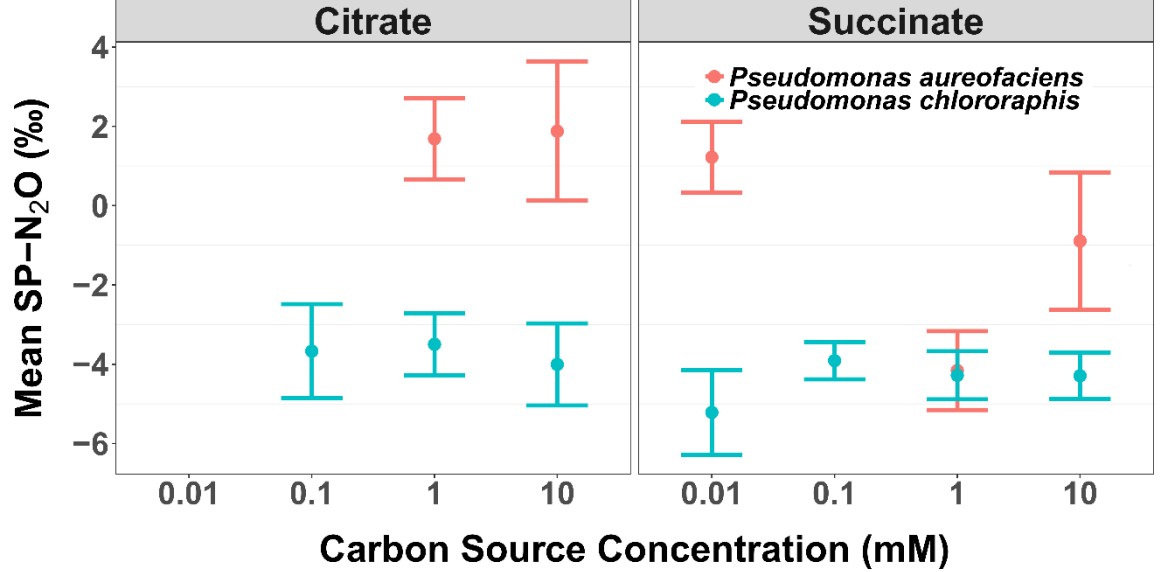

**Figure 4. The mean site preference (SP) of N$_2$O produced during the reduction of NO$_3^-$ by *Pseudomonas aureofaciens* (orange) and *Pseudomonas chlororaphis* (blue) with different concentrations of electron donors: citrate and succinate. Error bars indicate 1 standard deviation.**



**Figure 5. A schematic representation of the multi-step reduction of nitrate to nitrous oxide during denitrification specific to *Pseudomonas aureofaciens* and *Pseudomonas chlororaphis*, which lack the enzyme nitrous oxide reductase. The enzymes responsible for the reduction of nitrogen species appear in boxes with rounded corners and are indicated by three letter sequences: nitrate reductase (NAR), nitrite reductase (NIR), and nitric oxide reductase (NOR). A nitrate/nitrite transporter protein is presented as a hexagon. The position of the enzymes with respect to the periplasm, membrane, or cytoplasm identify the location of the enzymes in the cell. Vertical dashed arrows indicate diffusion of various nitrogen species into and out of the cell, and curved dashed arrows represent transport across the membrane. Solid arrows represent enzyme catalysed reduction steps.**