# Peer review of "Estimation of isotope variation of $N_2O$ during denitrification by *Pseudomonas aureofaciens* and *Pseudomonas chlororaphis*: Implications for $N_2O$ source apportionment"

_Biogeosciences, 2017_

## Referee Comment (RC1) · Anonymous Referee #1 · 21 Dec 2017

Summary

The authors present a well-conducted study of N2O production and isotope fractionation by two cultures of denitrifying bacteria during growth on two different carbon substrates. The authors measure the N and O isotopic composition, as well as the N2O intramolecular site preference, during production of N2O in an effort to better understand the underlying mechanisms that regulate its isotopic composition in the context of supply and type of organic carbon substrate and the multi-step process of denitrification. Because production of N2O during denitrification is a multi-step processes,

the authors observe marked variations in the N and O composition of product N2O over reaction progress. Because shifts in the apparent or net isotope effect during the course of the reaction implicitly violate the Rayleigh model for characterization of closed-system isotope dynamics, the authors adapt a clever new approach for determining the net isotope effects from the experimental data using an exponential function. Further, using probability density distributions, the authors illustrate how the net N and O isotope effects are interpreted to change over reaction progress for different species and carbon substrate types and concentrations. Importantly it is also demonstrated that N2O site preference remains a robust reflector of formation process – albeit with some notable inter-specific differences which remain enigmatic.

The paper is very well written, clear, concise, timely and insightful. I found it easy to read and appreciated the novel approach for determination of the net isotope effects and their probability density distributions. In particular this paper sheds important light on the fact that the isotopic composition of N2O (or any multi-step reaction intermediate, for that matter) can evolve in response to non-steady state reaction conditions, the build up of intermediate pools and other physiological controls on microbial metabolisms. I recommend publication of this manuscript after consideration of my comments detailed below.

General Comments:

1. I understand the authors' reluctance to over-interpret the $\delta$18O data given the fact that O isotope exchange between intermediates (notably NO2-) and water are known to occur. However, I do feel that more attention could be given to the $\delta$18O data. Certainly, no new experiments are needed (though parallel experiments in 18O labeled water would be insightful), but I am left wondering whether the authors too quickly neglect the consideration of these data by suggesting water O exchange plays such a large role in the data? More to the point, I wonder how the co-evolving $\delta$15N and $\delta$18O might be used to provide more insight, for example relating to carbon substrate concentrations and types? Is there any more information to be gained about water O isotope exchange

and thereby possibly the turnover of intermediate pools by closer consideration of these data in a more 'linked' fashion? Where there coherent trends in the $\delta$15N vs $\delta$18O that could be revealing? Also, were concentrations of NO2- measured during the sampling – in an effort to better constrain pool sizes of reaction intermediates? Even if the isotopic composition of NO2- was unknown – it might be useful for shedding light on variations of $\eta$18O.

2. Overall, I would appreciate a bit more insight on why the different carbon substrates might contribute to differential expression of net isotope effects. For example, how are citrate and succinate utilized by these two closely related organisms? Can the authors explain (even speculatively) about how these different carbon substrates might act to regulate expression of net isotope effects? This is an exciting and burgeoning avenue of research for microbial-isotope systematics across many elemental systems – and this study provides a unique perspective for denitrification, in particular. In general not enough attention was given to this result. Different carbon substrates were chosen – in part to explore such metabolic differences. What is the reader to learn from the experimental results using different carbon?

3. P9. The authors note that the N2O site preference is constant among treatments yet distinct between the two bacterial strains investigated. Towards offering some explanation for this distinction, they correctly suggest that the NOR step is the most critical (combination of two NO molecules to form N2O). However, it is unclear to me in this context how the fraction of NO remaining behind in the cell relates to the site preference (L 14). Site preference is conceptually thought to be the result of the combination of two NO molecules and to reflect the chemical (enzymatic) mechanisms by which this reaction occurs and is therefore agnostic to the composition of the precursor pool. As such – it is unclear to me how the NO precursor pool size (which may relate to its N isotopic composition) can play any role in the determination of site preference. Furthermore, it is stated that the N isotopic composition of the alpha and beta positions in the N2O molecule are 'factors related to site preference' – which makes little sense –

since these are exactly how site preference is calculated in the first place. Perhaps the authors are referring to the alpha and beta positions represented in the NO precursor molecules – which makes sense but should be clarified. Indeed if there is an argument to be made that the NO pool size somehow influences the partitioning among NO molecules destined for the alpha position from those destined for the beta position, this would be interesting and valuable to develop. At present, however, I am missing the point of this part of the discussion.

4. Figure 1. It would be helpful to know the composition of starting $NO_3^-$. Or alternatively are the Y-axes meant to reflect the difference between the starting $NO_3^-$ and the product $N_2O$? Figure 1 and 2 – while 'no positive values were calculated' – the distribution spills over into positive values in the upper panels of Figure 1 and all panels of Figure 2. It seems like the distribution was 'trimmed' for lower panels in Figure 1. I think some attention could be paid to addressing these differences – both in the text and in the figure caption. In particular – is there any reason to disregard positive values? Is the generation of a positive value in this context mathematically impossible? Figures 1 and 2 – are the tally marks meant to illustrate the distribution for each set of treatments (e.g., are there different color tally marks?). If so, I'm not sure I can distinguish among the different colors. It might be helpful to break out the different treatments and 'stack' the tally marks on top of one another?

Specific Comments:

P1 Ln 19 – Somewhat awkward to use this expression for the Rayleigh accumulated product without having definitions for the terms. Consider using 'accumulated product expression' instead perhaps?

P2 L10 – "include"

P5 L28 please define "HSD"

P5 L14 I realize that the coefficient 'b' is a simple fitting parameter, but I am wondering

if any sort of 'meaning' is discernible behind the absolute value of this coefficient? Can it be conceptualized as relating to some tangible aspect of the system?

---

## Referee Comment (RC2) · Anonymous Referee #2 · 24 Dec 2017

The manuscript by Joshua Haslun et al. provides the application of a Rayleigh approach to determine fractionation factors $\varepsilon$ for the multi-step $NO_3^-$ to $N_2O$ reduction by *Pseudomonas aureofaciens* and *Pseudomonas chlororaphis*. They observed a curvilinear relationship for $\delta^{15}N$-$N_2O$ and $\delta^{18}O$-$N_2O$ versus [–f lnf / (1-f)], which they attributed to the inter-play of fractionation factor of different enzymatic steps and diffusion of substrates and products. They provide a novel approach using non-linear least square regression to calculate net isotope effects (n) for the complete reaction progress (f = 1 to 0), which is of interest for the scientific community. In addition the study provides temporal trends for $N_2O$ SP, showing that SP is a robust indicator for specific production pathways.

The manuscript is very well written and thus easy to read and understand. There are no major weaknesses or shortcomings so I suggest publication after minor revisions.

General comments:

The novel approach to calculate the "net isotope effect" ($\eta$) is not very intuitive; therefore it should be rationalized how and why y = b x $e^{bx}$ (formula 5, Page 5 L14) is equivalent to $\delta^{15}N$-$N_2O$ – $\delta^{15}N$-$NO_3^-$ the standard approach to calculate the net isotope effect. In addition, results of the standard approach and the novel approach should be compared and discussed in the manuscript.

Specific comments:

Page 1 L17 ff: It should be mentioned here and elsewhere in the text that both bacterial strains lack the enzyme for $N_2O$ reduction as this might not be known by every reader of the manuscript.

Page 4 L5: The isotopic composition of the $NaNO_3$ should be given.

Page 4 L15: Which gas volume or range of volumes was sampled into the serum bottles?

Page 4 L26: The isotopic composition of the standards used for IRMS analysis of $\delta^{15}N^{\alpha}$, $\delta^{15}N^{\beta}$, $\delta^{15}N$ and $\delta^{18}O$-$N_2O$ should be given.

Page 4 L31: How was f determined? From the amount of substrate provided minus the cumulative amount of $N_2O$ produced? Please add the respective information here.

Page 6 section 3.1 and 3.2: Please provide results for $\eta^{15}N$ and $\eta^{18}O$ using the standard approach to calculate net isotope effects (e.g. $\delta^{15}N$-$N_2O$ – $\delta^{15}N$-$NO_3^-$).

Page 7 L5: One experiment with *Ps. aureofaciens* and succinate at 1 mM yields $N_2O$ with low SP similar to *Ps. chlororaphis*. How can this be explained or is it just an outlier?

Page 7 L12 – 21: This section, the application of the Rayleigh model, should be preferably placed in the introductory or method section.

Page 9 L14: The statement that SP depends on $\delta^{15}N^{\alpha}$ and $\delta^{15}N^{\beta}$ is trivial and could be omitted. The differences in SP between *Ps. aureofaciens* and *Ps. chlororaphis* are attributed to differences in the NO pool; which experiment could be used to check this hypothesis?

Page 8 (discussion): Temporal resolved data of $N_2O$ isotopic composition as used in this study could be preferably be collected using an online technique, e.g. laser spectroscopy – Please comment.

---

## Author Comment (AC1) · 2 Feb 2018

*The page numbers of the reviewer's responses link to the original document. The page numbers of manuscript changes refer the line numbers in the "track changed" document.

1. I understand the authors' reluctance to over-interpret the $\delta$18O data given the fact that O isotope exchange between intermediates (notably NO2-) and water are known to occur. However, I do feel that more attention could be given to the $\delta$18O data. Certainly,

no new experiments are needed (though parallel experiments in 18O labeled water would be insightful), but I am left wondering whether the authors too quickly neglect the consideration of these data by suggesting water O exchange plays such a large role in the data? More to the point, I wonder how the co-evolving $\delta$15N and $\delta$18O might be used to provide more insight, for example relating to carbon substrate concentrations and types? Is there any more information to be gained about water O isotope exchange and thereby possibly the turnover of intermediate pools by closer consideration of these data in a more 'linked' fashion? Where there coherent trends in the $\delta$15N vs $\delta$18O that could be revealing? Also, were concentrations of NO2- measured during the sampling – in an effort to better constrain pool sizes of reaction intermediates? Even if the isotopic composition of NO2- was unknown – it might be useful for shedding light on variations of $\eta$18O.

Response: To address the consideration that water O exchange plays a role in the data we graphically examined the covariation between $\delta$18O/$\delta$15N vs. –flnf/(1-f) as well as the covariation between $\delta$18O and $\delta$15N. Both covariation plots indicated that the relationship between $\delta$18O and $\delta$15N were similar among treatments and replicates. Therefore, no additional information can be gleaned by discussing the relationship between O and N isotope values. Additionally, the fact that we observed a kinetic isotope effect for $\delta$18O suggests that there is little exchange with H2O in the reaction vessels. We were unable to measure the concentration of NO2- in reaction vessels for two important reasons. First, sampling for NO2- and N2O would have added an incredible degree of complexity to the experiment, which could have led to inaccuracies and artefacts in the data. Second, additional punctures of the septa could contribute to N2O leakage into and out of the reaction vessels. Sampling this way would have doubled the number of punctures and thus increased the probability for N2O loss.

Manuscript Changes: P9 L6 – L8 – "Additionally visual inspection of the co-variation between $\delta$18O and $\delta$15N indicated similar trends among treatments and species, and the observed kinetic isotope effect for $\delta$18O suggests that there is little exchange with

H2O in the reaction vessels (Figure 1)."

2. Overall, I would appreciate a bit more insight on why the different carbon substrates might contribute to differential expression of net isotope effects. For example, how are citrate and succinate utilized by these two closely related organisms? Can the authors explain (even speculatively) about how these different carbon substrates might act to regulate expression of net isotope effects? This is an exciting and burgeoning avenue of research for microbial-isotope systematics across many elemental systems – and this study provides a unique perspective for denitrification, in particular. In general not enough attention was given to this result. Different carbon substrates were chosen – in part to explore such metabolic differences. What is the reader to learn from the experimental results using different carbon?

Response: We agree that understanding the influence of carbon-source on $\eta$ is important and timely. In fact, that was an initial objective of our research; however this objective became difficult to address when we observed that $\eta$ for $\delta15N$ and $\delta18O$ of N2O was not constant across the extent of the reaction. The fact that $\eta$ changes as the reaction progresses makes it difficult to statistically quantify (i.e. the sample size at a given point in the reaction) if the difference in $\eta$ between treatments occurred as a function of substrate. Moreover, we do not know which of the diffusive or enzymatic steps are controlling $\eta$ at a given extent of the reaction. To address the question regarding the influence of carbon-substrate on bulk isotope values, we will need to perform a detailed study that quantifies the isotope effects of the many nitrogen intermediates of denitrification simultaneously, a significant amount of work and therefore a study of its own.

Manuscript Changes: For the reasons outlined above, we do not feel that manuscript changes are necessary to respond to the comment.

3. The authors note that the N2O site preference is constant among treatments yet distinct between the two bacterial strains investigated. Towards offering some explanation

for this distinction, they correctly suggest that the NOR step is the most critical (combination of two NO molecules to form N2O). However, it is unclear to me in this context how the fraction of NO remaining behind in the cell relates to the site preference (L 14). Site preference is conceptually thought to be the result of the combination of two NO molecules and to reflect the chemical (enzymatic) mechanisms by which this reaction occurs and is therefore agnostic to the composition of the precursor pool. As such – it is unclear to me how the NO precursor pool size (which may relate to its N isotopic composition) can play any role in the determination of site preference. Furthermore, it is stated that the N isotopic composition of the alpha and beta positions in the N2O molecule are 'factors related to site preference' – which makes little sense since these are exactly how site preference is calculated in the first place. Perhaps the authors are referring to the alpha and beta positions represented in the NO precursor molecules – which makes sense but should be clarified. Indeed if there is an argument to be made that the NO pool size somehow influences the partitioning among NO molecules destined for the alpha position from those destined for the beta position, this would be interesting and valuable to develop. At present, however, I am missing the point of this part of the discussion.

Response: We addressed the issues above by altering the text that contributed to the lack of clarity. Please see the manuscript changes below.

Manuscript Changes: P10 L8 -P11 L11 - "In contrast to the results we observed for $\delta15N$ and $\delta18O$, isotopic discrimination was not evident for SP regardless of treatment (Figure 2). Instead, SP was constant during the course of the reaction. This finding is consistent with pure culture studies of nitrification and denitrification across multiple species (Frame and Casciotti, 2010; Sutka et al., 2003, 2006; Toyoda et al., 2005). The differences we observed in SP between species, however, is likely to relate to the factors that control SP. Unlike the case for bulk isotopes, SP is determined during a single reaction, the reduction of NO to N2O (Toyoda et al., 2005). Thus, as N2O reduction does not occur in P. aureofaciens or P. chlororaphis, SP is only influenced

by nitric oxide reductase (NOR) activity and diffusion of NO or N2O into or out of the cell. As SP is the difference between the $\delta$15N value of two N atoms that rely on the same NO substrate, SP is not dependent upon the isotopic composition of the initial substrate (Toyoda et al., 2005; Sutka et al., 2006). The observation that SP remained constant during bacterial denitrification, even though the extent of the reaction varied, (e.g. Sutka et al., 2006) suggests that the expressed fractionation for the $\alpha$ and $\beta$ N atoms during NO reduction were the same. If so, then one hypothesis is that f can vary markedly and SP will be constant. However, during production of N2O by pure fungal cytochrome P450 NOR enzyme, distinct fractionation factors for the $\alpha$ and $\beta$ N atoms were observed and it was proposed that observations of constant SP values during production by fungi were the result of f, or the internal pool size of NO, being held relatively constant during cellular metabolism (Yang et al., 2014). We observed a minor but significant different in SP between two species of Pseudomonas sp. during N2O production that is consistent with a difference in the internal pool size of NO within the cell. The abundance of NO within the cell will depend on its production, reduction, and losses due to diffusion into or out of the cell, all of which could vary between species. We do not know, for example, the degree to which the rate of NO production intrinsically differs between the cd1-type NIR of P. chlororaphis and copper containing NIR of P. aureofaciens or how gene expression may alter these rates. We posit that small differences in SP between and even within species in our study and others may relate to the size of the NO pool available to NOR."

4. Figure 1. It would be helpful to know the composition of starting NO3-. Or alternatively are the Y-axes meant to reflect the difference between the starting NO3- and the product N2O? Figure 1 and 2 – while 'no positive values were calculated' – the distribution spills over into positive values in the upper panels of Figure 1 and all panels of Figure 2. It seems like the distribution was 'trimmed' for lower panels in Figure 1. I think some attention could be paid to addressing these differences – both in the text and in the figure caption. In particular – is there any reason to disregard positive values? Is the generation of a positive value in this context mathematically impossible? Figures 1

and 2 – are the tally marks meant to illustrate the distribution for each set of treatments (e.g., are there different color tally marks?). If so, I'm not sure I can distinguish among the different colors. It might be helpful to break out the different treatments and 'stack' the tally marks on top of one another?

Response: We added the isotopic composition of the nitrate source to the text. See below for the manuscript changes. We felt that it would be helpful to review the basis of the generation of the density distributions from our estimates of $\eta$ in order to address the reviewer's concerns regarding the density distributions. As stated in the text, we applied Gaussian kernel density estimation to determine the density distribution of $\eta$ predicted to occur across the full extent of the reactions. Kernel density estimation is a non-parametric method of determining the probability density function of a random continuous variable. The kernel applies a density function to each point of data. A probability density function is then created by adding up the sum of functions for each of the supplied points and dividing by the number of data. The output is an estimate of the relative densities of values across the range of $\eta$. For example, the density distribution predicts that for P. aureofaciens provided 10 mM citrate, a $\eta$ value of -100 ‰ would occur relatively infrequently. The horizontal arrow on the x-axis of the graphs indicates that a value of large magnitude, such as the $\eta$ = -100 ‰ previously described would be observed at the beginning of the reaction. The tick marks are the $\eta$ values estimated from the derivative of the non-linear model. These $\eta$ values were used to construct the density distribution. These tick marks allow one to compare the estimated $\eta$ values to the estimated density distribution. This discussion is reflected by changes in the text outlined below. The reviewer's argument regarding positive values is important. If we examine the curves in figure 1, we see that each curve has a negative slope over the course of the reaction, indicative of a normal isotope effect when the x-axis variable is $-f\ln f/(1-f)$. If we draw our attention to the $\delta$15N isotope values for P. chlororaphis supplied with 10 mM citrate, we note that the left-hand side of the curve is approaching an asymptote with a slope approaching 0. Transition to a positive slope would require that $\delta$15N isotope values became more negative. This would indicate that an inverse

isotope effect is contributing fractionation, something that is not supported by our data. Moreover, because P. aureofaciens and P. chlororaphis do not produce nitrous oxide reductase it is unlikely that an inverse isotope effect would be observed. Therefore, we have deliberately decided to cut off the density distributions at $\eta = 0$ ‰ and have revised figure 2 and 3 accordingly.

Manuscript Changes: P4 L11 – "The $\delta$15N and $\delta$18O of the NO3- source was 5.4 ‰ and 24.4 ‰ respectively." P6 L2-4 – "Values of "a" affect the y-intercept with larger values contributing to increased prediction of the final isotope value of the reaction. Values of "b" affect the rate of change of the isotope values particularly at the beginning of the reaction. Larger values of "b" result in a more gradual rate of change, whereas as lower values of "b" increase the initial slope. P6 L12-16 – "We used kernel density estimation to illustrate the density distribution (DD) of $\eta$ across the extent of the reaction observed. Kernel density estimation is a non-parametric method of determining the probability density function of a random continuous variable. Probability density functions were determined with a Gaussian smoothing kernel from 50 equally spaced estimates of $\eta$ spanning the complete extent of the reaction (i.e. f = 0 to 1). The bandwidth was set to 1 for each density estimate." P7 L3 - L6 – "Values of $\eta$15N greater than 0 were not observed. Such values would indicate that an inverse isotope effect is contributing fractionation, something that is not indicated by our data. Moreover, because P. aureofaciens and P. chlororaphis do not produce nitrous oxide reductase it is unlikely that an inverse isotope effect would be observed. Therefore, we have deliberately decided to cut off the density distributions at $\eta = 0$ ‰." P16-21 - The lines as well as tick marks for figures 2 and 3 have been changed to make comparisons of the treatments easier. The x-axes have been changed to reflect that $\eta$ values greater than zero were not observed in our reactions. The term PDDs in text and in figure legends has been changed to density distributions (DD) in text to reflect the previous in text changes."

Specific Comments-

P1 L19: Somewhat awkward to use this expression for the Rayleigh accumulated product without having definitions for the terms. Consider using 'accumulated product expression' instead perhaps?

Response: We will change this to "the extent of the reaction." for the abstract. We believe that there is some virtue in stating the fraction of the accumulated product for the rest of the manuscript.

Manuscript Change: P1 L20-21 – Changed the expression to "the extent of the reaction".

P2 L10: "include"

Response: Changed "includes" to "include" as recommended.

Manuscript Change: P2 L15 – "includes" changed to "include"

P5 L28: please define "HSD"

Response: We remove HSD and include the full term in text.

Manuscript Change: P6 L22 – Changed "HSD" to "honest significant difference test"

P5 L14: I realize that the coefficient 'b' is a simple fitting parameter, but I am wondering if any sort of 'meaning' is discernible behind the absolute value of this coefficient? Can it be conceptualized as relating to some tangible aspect of the system?

Response: The coefficient "b" affects the rate of change of the isotope value for a given non-linear function closer to the onset of the reaction. Increased values of "b" produce a more gradual rate of change, whereas lower "b" values increase the rate of change of isotope values producing a very steep initial slope. We have included text to explain this effect.

Manuscript Change: P6 L3-4 - "Values of "a" affect the y-intercept with larger values contributing to increased prediction of the final isotope value of the reaction. Values of "b" affect the rate of change of the isotope values particularly at the beginning of

the reaction. Larger values of "b" result in a more gradual rate of change, whereas as lower values of "b" increase the initial slope."

Please also note the supplement to this comment:
https://www.biogeosciences-discuss.net/bg-2017-463/bg-2017-463-AC1-supplement.pdf

[Figure]

**Fig. 1.** Density distributions (DD) of $\delta$15N net isotope effects ($\eta$)

[Figure]

**Fig. 2.** Density distributions (DD) of $\delta 18O$ net isotope effects ($\eta$)

**Supplement:**

[revised manuscript text omitted]

**Commented [JH7]:** Reviewer #1: P5L14 Please see the comment I have written in the RefereeResponses document

**Commented [JH8]:** Reviewer #1: Specific comment P5L28

*aureofaciens* ranged from -77.5 ‰ to -18.4 ‰ and -106.2 ‰ to -11.4 ‰ for citrate and succinate, respectively, while values ranged from -119 ‰ to -9.2 ‰ and -82.1 ‰ to -5.1 ‰ for citrate and succinate, respectively, during denitrification by *P. chlororaphis* (Figure 2, Supplementary Table 1). Values of $\eta^{15}N$ greater than 0 were not observed. Such values would indicate that an inverse isotope effect is contributing fractionation, something that is not indicated by our data. Moreover, because *P. aureofaciens* and *P. chlororaphis* do not produce nitrous oxide reductase it is unlikely that an inverse isotope effect would be observed. Therefore, we have deliberately decided to cut off the density distributions at $\eta = 0$ ‰. 
[revised manuscript text omitted]

---

## Author Comment (AC2) · 2 Feb 2018

*The page numbers of the reviewer's responses link to the original document. The page numbers of manuscript changes refer the line numbers in the "track changed" document not the corrected final version.

1. The novel approach to calculate the "net isotope effect" (Æđ) is not very intuitive; therefore it should be rationalized how and why y = b x ebx (formula 5, Page 5 L14) is equivalent to $\delta15N$-N2O – $\delta15N$-NO3- the standard approach to calculate the net

isotope effect. In addition, results of the standard approach and the novel approach should be compared and discussed in the manuscript.

Response: The reviewer refers to the $\Delta$ as the common approach to calculate a NIE ($\eta$). Capital delta is an approximation of the $\eta$, the slope of the Rayleigh plot, and therefore additional error is associated with this particular calculation of $\eta$ (please see O'Neil 1986). Rayleigh models have been traditionally applied to describe isotope effects including $\eta$. The curvilinear behaviour of our data demonstrates that we violate the linearity, a critical assumption of the application of Rayleigh models to the estimation of isotopic fractionation. Moreover, a single value for $\eta$ cannot describe our data. Thus, we provide an equation for $\eta$ that allows one to obtain an estimate of the isotope effect at any point in the reaction, rather than producing a single value that would fall short of describing our curvilinear system. O'Neil, J.R. (1986) Theoretical and experimental aspects of isotopic fractionation. In Stable Isotopes in High Temperature Geological Processes (eds. J.W. Valley, H.P. Taylor, Jr., and J.R. O'Neil) Rev. Mineral., 16, 1-40

Manuscript Changes: We feel that the response above addresses the concerns of the reviewer.

Specific Comments-

P1 L17: It should be mentioned here and elsewhere in the text that both bacterial strains lack the enzyme for N2O reduction as this might not be known by every reader of the manuscript.

Response: We made changes in the text on page 1 to address the reviewer's concerns.

Manuscript Change: P1 L17-18 – The sentence "Pseudomonas aureofaciens and P. chlororaphis lack the gene nitrous oxide reductase, NosZ, and therefore N2O is the terminal product of the reduction of NO3-." has been included. Additionally, the sentence at the end of the introduction, P3 L6, addresses the lack of N2O reductase in the two species utilized in this study.

P4 L5: The isotopic composition of the NaNO3 should be given.

Response: We included the requested values in text.

Manuscript Change: P4 L11 - "The $\delta$15N and $\delta$18O of the NO3- source was 5.4‰ and 24.4‰ respectively."

P4 L15: Which gas volume or range of volumes was sampled into the serum bottles?

Response: We addressed this issue in text. Please see the changes below.

Manuscript Change: P4 L20-22 - "Headspace samples between 200 $\mu$L and 500 $\mu$L of each of the 3 cultures were injected into 60 ml serum bottles (one per culture) that had been sparged with UHP N2 for 15 min, and stored for isotope analysis. Each bottle contained between 5 nmols and 15 nmols of N2O for isotopic analysis."

P4 L26: The isotopic composition of the standards used for IRMS analysis of $\delta$15N$\alpha$, $\delta$15N$\beta$, $\delta$15N and $\delta$18O-N2O should be given.

Response: We have included a paragraph in the methods section that provides the values of standards used for analysis.

Manuscript Changes: P4 L28- P5 L2 - "Our internal laboratory pure N2O tank standard (MSU Tank B) was isotopically characterized by analysis relative to the USGS51 and USGS52 reference materials (https://isotopes.usgs.gov/lab/referencematerials.html). Following the guidelines proposed by Coplen (2011), we report here the isotope values of the reference materials as well as our internal laboratory standard. The $\delta$15N, $\delta$18O, $\delta$15N$\alpha$, $\delta$15N$\beta$, and SP values of USGS51 and USGS52 are 1.32 ‰ 41.23 ‰ 0.48 ‰ 2.15 ‰ and -1.67 ‰ and 0.44 ‰ 40.64, 13.52, -12.64 ‰ and 26.15 ‰ respectively. The $\delta$15N, $\delta$18O, $\delta$15N$\alpha$, $\delta$15N$\beta$, and SP values of reference MSU Tank C are -0.9 ‰ 0.7 ‰ -2.6 ‰ 39.6 ‰ and 3.4 ‰ respectively. The $\delta$15N, $\delta$18O, $\delta$15N$\alpha$, $\delta$15N$\beta$, and SP values of the isotope standard MSU Tank B are -0.5 ‰ 11.13 ‰ -12.2 ‰ 40.8 ‰ and 23.3 ‰ respectively. All nitrogen isotope values are reported with respect to the international Air-N2 standard, and all oxygen isotope values with respect to VSMOW."

P4 L31: How was f determined? From the amount of substrate provided minus the cumulative amount of N2O produced? Please add the respective information here.

Response: We have included our mathematical approach to determining f.

Manuscript Change: P5 L23-24 - "The fraction of substrate remaining was determined by dividing twice the amount of N2O produced by the total amount of nitrate added, and then subtracting this quantity from 1."

P6 section 3.1 and 3.2: Please provide results for Æđ15N and Æđ18O using the standard approach to calculate net isotope effects (e.g. $\delta$15N-N2O $- \delta$15N-NO3-).

Response: Please see the response to Reviewer #1's general comments 1 and 4 for an explanation of the changes made to the manuscript.

Manuscript Change: Please see the previous manuscript changes made as recommended in the response above.

P7 L5: One experiment with P. aureofaciens and succinate at 1 mM yields N2O with low SP similar to P. chlororaphis. How can this be explained or is it just an outlier?

Response: We have changed the text to make the interpretation of our results reflect the reviewer's concerns.

Manuscript Change: P7 L30-32 – The text has been changed to read "In four of the treatments, the average SP of P. chlororaphis denitrification was lower than that of P. aureofaciens. This resulted in a difference of 4.1‰ between the average SP of P. chlororaphis and P. aureofaciens."

P7 L12 – 21: This section, the application of the Rayleigh model, should be preferably placed in the introductory or method section.

Response: As recommended by Reviewer #2, we have moved the application of the Rayleigh model to the method section. Please see the text and below for the precise changes.

Manuscript Changes: P5 L12-22 - "According to convention (Mariotti et al., 1981), the magnitude of the isotopic fractionation factor ($\alpha$) for a single unidirectional reaction is defined by the rate constants of the light (k1) and heavy (k2) isotopically substituted compounds: $\alpha = k_2/k_1$. (3) Further, the isotopic enrichment factor, $\varepsilon$, is defined as $\varepsilon = (\alpha-1) \times 1000$, (4) and can be estimated from the slope of the linear relationship described by the Rayleigh model: $\delta^{15}N_p = \delta^{15}N_{so} - \varepsilon_{(p/s)} [((f \ln f)/((1-f))]$; (5) where $\delta 15 N_p$ is the isotope value of the accumulated product, $\delta 15 N_{so}$ is the isotope value of the initial substrate, $\varepsilon$ is the fractionation factor, and f is the fraction of substrate remaining (Mariotti et al., 1981)."

P9 L14: The statement that SP depends on $\delta 15 N\alpha$ and $\delta 15 N\beta$ is trivial and could be omitted. The differences in SP between Ps. aureofaciens and Ps. chlororaphis are attributed to differences in the NO pool; which experiment could be used to check this hypothesis?

Response: We have changed the text to reflect that these values are used to calculate SP rather than SP being related to them. As far as designing an experiment to determine if SP is affected by the NO pool, this is an inherently difficult problem, which arises from the difficulty in determining the concentration of NO available to the bacterium within the periplasmic space. We are currently beginning to examine NO reduction during enzymatic reduction both in vitro and in vivo.

Manuscript Changes: See the response for Reviewer #1 comment 3; P10 L8 – P11 L11.

Page 8 (discussion): Temporal resolved data of N2O isotopic composition as used in this study could be preferably be collected using an online technique, e.g. laser spectroscopy – Please comment.

Response: While online techniques such as off-axis integrated cavity output spectroscopy (OA-ICOS) and cavity ring down spectroscopy (CRDS) offer the capability to temporally resolve changes in N2O isotopomer values, the techniques have unique

and challenges inherent to the technology to overcome. For instance, many obstacles remain that constrain the field, mesocosm, and in vitro application of OA-ICOS and CRDS including production and standardization of reference gases spanning a range of SP values and N2O concentrations, as well as the standardization of methods for data analysis. Among other labs, our group is currently working to address challenges such as these as on-line measurements promise rapid continuous field-based and mesocosm data necessary for managing and monitoring N2O flux at local, ecosystem, and global scales. However, these topics have been reviewed and discusses in other peer-reviewed journals (Ostrom and Ostrom 2017). Therefore we feel that including such information detracts from the overall message we present in this manuscript. Ostrom, N.E., Ostrom, P.H., 2017. Mining the isotopic complexity of nitrous oxide: a review of challenges and opportunities. Biogeochemistry, 132:3, 359-372.

Manuscript Changes: Based upon the response above we do not feel that an in text response to the comment is necessary.

Please also note the supplement to this comment:
https://www.biogeosciences-discuss.net/bg-2017-463/bg-2017-463-AC2-supplement.pdf

**Supplement:**

[revised manuscript text omitted]

**Commented [JH7]:** Reviewer #1: P5L14 Please see the comment I have written in the RefereeResponses document

**Commented [JH8]:** Reviewer #1: Specific comment P5L28

*aureofaciens* ranged from -77.5 ‰ to -18.4 ‰ and -106.2 ‰ to -11.4 ‰ for citrate and succinate, respectively, while values ranged from -119 ‰ to -9.2 ‰ and -82.1 ‰ to -5.1 ‰ for citrate and succinate, respectively, during denitrification by *P. chlororaphis* (Figure 2, Supplementary Table 1). Values of $\eta^{15}N$ greater than 0 were not observed. Such values would indicate that an inverse isotope effect is contributing fractionation, something that is not indicated by our data. Moreover, because *P. aureofaciens* and *P. chlororaphis* do not produce nitrous oxide reductase it is unlikely that an inverse isotope effect would be observed. Therefore, we have deliberately decided to cut off the density distributions at $\eta = 0$ ‰. 
[revised manuscript text omitted]

---

## Author Response (AR1)

March 18, 2017

Ximing Wang
Associate Editor
Biogeosciences
Göttingen, Germany

Dear Ximing Wang,

Please find the corrected pdf of our manuscript, titled "Estimation of isotope variation of $N_2O$ during denitrification by *Pseudomonas aureofaciens* and *Pseudomonas chlororaphis*: Implications for $N_2O$ source apportionment". We have provided the corrected version from the previous reviews as there were no other requested changes available. We thank you for working with us during the review process.

Sincerely,

Joshua A Haslun

---

## Author Response (AR2)

*The page numbers of the reviewer's responses link to the original document. The page numbers of manuscript changes refer the line numbers in the "track changed" document not the corrected final version.

**Anonymous Referee #1**

**General Comments-**

1. I understand the authors' reluctance to over-interpret the $\delta^{18}O$ data given the fact that O isotope exchange between intermediates (notably $NO_2^-$) and water are known to occur. However, I do feel that more attention could be given to the $\delta^{18}O$ data. Certainly, no new experiments are needed (though parallel experiments in $^{18}O$ labeled water would be insightful), but I am left wondering whether the authors too quickly neglect the consideration of these data by suggesting water O exchange plays such a large role in the data? More to the point, I wonder how the co-evolving $\delta^{15}N$ and $\delta^{18}O$ might be used to provide more insight, for example relating to carbon substrate concentrations and types? Is there any more information to be gained about water O isotope exchange and thereby possibly the turnover of intermediate pools by closer consideration of these data in a more 'linked' fashion? Where there coherent trends in the $\delta^{15}N$ vs $\delta^{18}O$ that could be revealing? Also, were concentrations of $NO_2^-$ measured during the sampling – in an effort to better constrain pool sizes of reaction intermediates? Even if the isotopic composition of $NO_2^-$ was unknown – it might be useful for shedding light on variations of $\eta^{18}O$.

Response:

To address the consideration that water O exchange plays a role in the data we graphically examined the covariation between $\delta^{18}O/\delta^{15}N$ vs. $-flnf/(1-f)$ as well as the covariation between $\delta^{18}O$ and $\delta^{15}N$. Both covariation plots indicated that the relationship between $\delta^{18}O$ and $\delta^{15}N$ were similar among treatments and replicates. Therefore, no additional information can be gleaned by discussing the relationship between O and N isotope values. Additionally, the fact that we observed a kinetic isotope effect for $\delta^{18}O$ suggests that there is little exchange with $H_2O$ in the reaction vessels.

We were unable to measure the concentration of $NO_2^-$ in reaction vessels for two important reasons. First, sampling for $NO_2^-$ and $N_2O$ would have added an incredible degree of complexity to the experiment, which could have led to

inaccuracies and artefacts in the data. Second, additional punctures of the septa could contribute to $N_2O$ leakage into and out of the reaction vessels. Sampling this way would have doubled the number of punctures and thus increased the probability for $N_2O$ loss.

Manuscript Changes:

P9 L6 – L8 – "Additionally visual inspection of the co-variation between $\delta^{18}O$ and $\delta^{15}N$ indicated similar trends among treatments and species, and the observed kinetic isotope effect for $\delta^{18}O$ suggests that there is little exchange with $H_2O$ in the reaction vessels (Figure 1)."

2. Overall, I would appreciate a bit more insight on why the different carbon substrates might contribute to differential expression of net isotope effects. For example, how are citrate and succinate utilized by these two closely related organisms? Can the authors explain (even speculatively) about how these different carbon substrates might act to regulate expression of net isotope effects? This is an exciting and burgeoning avenue of research for microbial-isotope systematics across many elemental systems – and this study provides a unique perspective for denitrification, in particular. In general not enough attention was given to this result. Different carbon substrates were chosen – in part to explore such metabolic differences. What is the reader to learn from the experimental results using different carbon?

Response:

We agree that understanding the influence of carbon-source on $\eta$ is important and timely. In fact, that was an initial objective of our research; however this objective became difficult to address when we observed that $\eta$ for $\delta^{15}N$ and $\delta^{18}O$ of $N_2O$ was not constant across the extent of the reaction. The fact that $\eta$ changes as the reaction progresses makes it difficult to statistically quantify (i.e. the sample size at a given point in the reaction) if the difference in $\eta$ between treatments occurred as a function of substrate. Moreover, we do not know which of the diffusive or enzymatic steps are controlling $\eta$ at a given extent of the reaction. To address the question regarding the influence of carbon-substrate on bulk isotope values, we will need to perform a detailed study that quantifies the isotope effects of the many nitrogen intermediates of denitrification simultaneously, a significant amount of work and therefore a study of its own.

Manuscript Changes:

For the reasons outlined above, we do not feel that manuscript changes are necessary to respond to the comment.

3. The authors note that the $N_2O$ site preference is constant among treatments yet distinct between the two bacterial strains investigated. Towards offering some explanation for this distinction, they correctly suggest that the NOR step is the most critical (combination of two NO molecules to form $N_2O$). However, it is unclear to me in this context how the fraction of NO remaining behind in the cell relates to the site preference (L 14). Site preference is conceptually thought to be the result of the combination of two NO molecules and to reflect the chemical (enzymatic) mechanisms by which this reaction occurs and is therefore agnostic to the composition of the precursor pool. As such – it is unclear to me how the NO precursor pool size (which may relate to its N isotopic composition) can play any role in the determination of site preference. Furthermore, it is stated that the N isotopic composition of the alpha and beta positions in the $N_2O$ molecule are 'factors related to site preference' – which makes little sense since these are exactly how site preference is calculated in the first place. Perhaps the authors are referring to the alpha and beta positions represented in the NO precursor molecules – which makes sense but should be clarified. Indeed if there is an argument to be made that the NO pool size somehow influences the partitioning among NO molecules destined for the alpha position from those destined for the beta position, this would be interesting and valuable to develop. At present, however, I am missing the point of this part of the discussion.

Response:

We addressed the issues above by altering the text that contributed to the lack of clarity. Please see the manuscript changes below.

Manuscript Changes:

P10 L8 -P11 L11 - "In contrast to the results we observed for $\delta^{15}N$ and $\delta^{18}O$, isotopic discrimination was not evident for SP regardless of treatment (Figure 2). Instead, SP was constant during the course of the reaction. This finding is consistent with pure culture studies of nitrification and denitrification across multiple species (Frame and Casciotti, 2010; Sutka et al., 2003, 2006; Toyoda et al., 2005). The differences we observed in SP between species, however, is likely to relate to the factors that control SP. Unlike the case for bulk isotopes, SP is determined during a single reaction, the reduction of NO to $N_2O$ (Toyoda et al., 2005). Thus, as $N_2O$ reduction does not occur in *P.*

*aureofaciens* or *P. chlororaphis*, SP is only influenced by nitric oxide reductase (NOR) activity and diffusion of NO or $N_2O$ into or out of the cell. As SP is the difference between the $\delta^{15}N$ value of two N atoms that rely on the same NO substrate, SP is not dependent upon the isotopic composition of the initial substrate (Toyoda et al., 2005; Sutka et al., 2006). The observation that SP remained constant during bacterial denitrification, even though the extent of the reaction varied, (e.g. Sutka et al., 2006) suggests that the expressed fractionation for the α and β N atoms during NO reduction were the same. If so, then one hypothesis is that f can vary markedly and SP will be constant. However, during production of $N_2O$ by pure fungal cytochrome P450 NOR enzyme, distinct fractionation factors for the α and β N atoms were observed and it was proposed that observations of constant SP values during production by fungi were the result of f, or the internal pool size of NO, being held relatively constant during cellular metabolism (Yang et al., 2014). We observed a minor but significant different in SP between two species of *Pseudomonas* sp. during $N_2O$ production that is consistent with a difference in the internal pool size of NO within the cell. The abundance of NO within the cell will depend on its production, reduction, and losses due to diffusion into or out of the cell, all of which could vary between species. We do not know, for example, the degree to which the rate of NO production intrinsically differs between the $cd_1$-type NIR of *P. chlororaphis* and copper containing NIR of *P. aureofaciens* or how gene expression may alter these rates. We posit that small differences in SP between and even within species in our study and others may relate to the size of the NO pool available to NOR."

4. Figure 1. It would be helpful to know the composition of starting $NO_3^-$. Or alternatively are the Y-axes meant to reflect the difference between the starting $NO_3^-$ and the product $N_2O$? Figure 1 and 2 – while 'no positive values were calculated' – the distribution spills over into positive values in the upper panels of Figure 1 and all panels of Figure 2. It seems like the distribution was 'trimmed' for lower panels in Figure 1. I think some attention could be paid to addressing these differences – both in the text and in the figure caption. In particular – is there any reason to disregard positive values? Is the generation of a positive value in this context mathematically impossible? Figures 1 and 2 – are the tally marks meant to illustrate the distribution for each set of treatments (e.g., are there different color tally marks?). If so, I'm not sure I can distinguish among the different colors. It might be helpful to break out the different treatments and 'stack' the tally marks on top of one another?

Response:

We added the isotopic composition of the nitrate source to the text. See below for the manuscript changes. We felt that it would be helpful to review the basis of the generation of the density distributions from our estimates of $\eta$ in order to address the reviewer's concerns regarding the density distributions. As stated in the text, we applied Gaussian kernel density estimation to determine the density distribution of $\eta$ predicted to occur across the full extent of the reactions. Kernel density estimation is a non-parametric method of determining the probability density function of a random continuous variable. The kernel applies a density function to each point of data. A probability density function is then created by adding up the sum of functions for each of the supplied points and dividing by the number of data. The output is an estimate of the relative densities of values across the range of $\eta$. For example, the density distribution predicts that for *P. aureofaciens* provided 10 mM citrate, a $\eta$ value of -100 ‰ would occur relatively infrequently. The horizontal arrow on the x-axis of the graphs indicates that a value of large magnitude, such as the $\eta$ = -100 ‰ previously described would be observed at the beginning of the reaction. The tick marks are the $\eta$ values estimated from the derivative of the non-linear model. These $\eta$ values were used to construct the density distribution. These tick marks allow one to compare the estimated $\eta$ values to the estimated density distribution. This discussion is reflected by changes in the text outlined below.

The reviewer's argument regarding positive values is important. If we examine the curves in figure 1, we see that each curve has a negative slope over the course of the reaction, indicative of a normal isotope effect when the x-axis variable is $-f\ln f/(1-f)$. If we draw our attention to the $\delta^{15}N$ isotope values for *P. chlororaphis* supplied with 10 mM citrate, we note that the left-hand side of the curve is approaching an asymptote with a slope approaching 0. Transition to a positive slope would require that $\delta^{15}N$ isotope values became more negative. This would indicate that an inverse isotope effect is contributing fractionation, something that is not supported by our data. Moreover, because *P. aureofaciens* and *P. chlororaphis* do not produce nitrous oxide reductase it is unlikely that an inverse isotope effect would be observed. Therefore, we have deliberately decided to cut off the density distributions at $\eta$ = 0 ‰ and have revised figure 2 and 3 accordingly.

Manuscript Changes:

P4 L11 – "The $\delta^{15}N$ and $\delta^{18}O$ of the $NO_3^-$ source was 5.4 ‰ and 24.4 ‰, respectively."

P6 L2-4 – "Values of "a" affect the y-intercept with larger values contributing to increased prediction of the final isotope value of the reaction. Values of "b" affect the rate of change of the isotope values particularly at the

beginning of the reaction. Larger values of "b" result in a more gradual rate of change, whereas as lower values of "b" increase the initial slope.

P6 L12-16 – "We used kernel density estimation to illustrate the density distribution (DD) of η across the extent of the reaction observed. Kernel density estimation is a non-parametric method of determining the probability density function of a random continuous variable. Probability density functions were determined with a Gaussian smoothing kernel from 50 equally spaced estimates of η spanning the complete extent of the reaction (i.e. f = 0 to 1). The bandwidth was set to 1 for each density estimate."

P7 L3 - L6 – "Values of $\eta^{15}N$ greater than 0 were not observed. Such values would indicate that an inverse isotope effect is contributing fractionation, something that is not indicated by our data. Moreover, because *P. aureofaciens* and *P. chlororaphis* do not produce nitrous oxide reductase it is unlikely that an inverse isotope effect would be observed. Therefore, we have deliberately decided to cut off the density distributions at η = 0 ‰."

P16-21 - The lines as well as tick marks for figures 2 and 3 have been changed to make comparisons of the treatments easier. The x-axes have been changed to reflect that η values greater than zero were not observed in our reactions. The term PDDs in text and in figure legends has been changed to density distributions (DD) in text to reflect the previous in text changes."

**Specific Comments-**

P1 L19: Somewhat awkward to use this expression for the Rayleigh accumulated product without having definitions for the terms. Consider using 'accumulated product expression' instead perhaps?

Response:

We will change this to "the extent of the reaction." for the abstract. We believe that there is some virtue in stating the fraction of the accumulated product for the rest of the manuscript.

Manuscript Change:

P1 L20-21 – Changed the expression to "the extent of the reaction".

P2 L10: "include"

Response:

Changed "includes" to "include" as recommended.

Manuscript Change:

P2 L15 – "includes" changed to "include"

P5 L28: please define "HSD"

Response:

We remove HSD and include the full term in text.

Manuscript Change:

P6 L22 – Changed "HSD" to "honest significant difference test"

P5 L14: I realize that the coefficient 'b' is a simple fitting parameter, but I am wondering if any sort of 'meaning' is discernible behind the absolute value of this coefficient? Can it be conceptualized as relating to some tangible aspect of the system?

Response:

The coefficient "b" affects the rate of change of the isotope value for a given non-linear function closer to the onset of the reaction. Increased values of "b" produce a more gradual rate of change, whereas lower "b" values increase the rate of change of isotope values producing a very steep initial slope. We have included text to explain this effect.

Manuscript Change:

P6 L3-4 - "Values of "a" affect the y-intercept with larger values contributing to increased prediction of the final isotope value of the reaction. Values of "b" affect the rate of change of the isotope values particularly at the beginning of the reaction. Larger values of "b" result in a more gradual rate of change, whereas as lower values of "b" increase the initial slope."

**Anonymous Referee #2**

**General Comments-**

1.  The novel approach to calculate the "net isotope effect" ($\eta$) is not very intuitive; therefore it should be rationalized how and why $y = b \times e^{bx}$ (formula 5, Page 5 L14) is equivalent to $\delta^{15}N\text{-}N_2O – \delta^{15}N\text{-}NO_3^-$ the standard approach to calculate the net isotope effect. In addition, results of the standard approach and the novel approach should be compared and discussed in the manuscript.

Response:

The reviewer refers to the $\Delta$ as the common approach to calculate a NIE ($\eta$). Capital delta is an approximation of the $\eta$, the slope of the Rayleigh plot, and therefore additional error is associated with this particular calculation of $\eta$ (please see O'Neil 1986). Rayleigh models have been traditionally applied to describe isotope effects including $\eta$. The curvilinear behaviour of our data demonstrates that we violate the linearity, a critical assumption of the application of Rayleigh models to the estimation of isotopic fractionation. Moreover, a single value for $\eta$ cannot describe our data. Thus, we provide an equation for $\eta$ that allows one to obtain an estimate of the isotope effect at any point in the reaction, rather than producing a single value that would fall short of describing our curvilinear system.

O'Neil, J.R. (1986) Theoretical and experimental aspects of isotopic fractionation. In Stable Isotopes in High Temperature Geological Processes (eds. J.W. Valley, H.P. Taylor, Jr., and J.R. O'Neil) Rev. Mineral., 16, 1-40

Manuscript Changes:

We feel that the response above addresses the concerns of the reviewer.

**Specific Comments-**

P1 L17: It should be mentioned here and elsewhere in the text that both bacterial strains lack the enzyme for $N_2O$ reduction as this might not be known by every reader of the manuscript.

Response:

We made changes in the text on page 1 to address the reviewer's concerns.

Manuscript Change:

P1 L17-18 – The sentence "*Pseudomonas aureofaciens* and *P. chlororaphis* lack the gene nitrous oxide reductase, NosZ, and therefore $N_2O$ is the terminal product of the reduction of $NO_3^-$." has been included. Additionally, the sentence at the end of the introduction, P3 L6, addresses the lack of $N_2O$ reductase in the two species utilized in this study.

P4 L5: The isotopic composition of the $NaNO_3$ should be given.

Response:

We included the requested values in text.

Manuscript Change:

P4 L11 - "The $\delta^{15}N$ and $\delta^{18}O$ of the $NO_3^-$ source was 5.4‰ and 24.4‰, respectively."

P4 L15: Which gas volume or range of volumes was sampled into the serum bottles?

Response:

We addressed this issue in text. Please see the changes below.

Manuscript Change:

P4 L20-22 - "Headspace samples between 200 μL and 500 μL of each of the 3 cultures were injected into 60 ml serum bottles (one per culture) that had been sparged with UHP $N_2$ for 15 min, and stored for isotope analysis. Each bottle contained between 5 nmols and 15 nmols of $N_2O$ for isotopic analysis."

P4 L26: The isotopic composition of the standards used for IRMS analysis of $\delta^{15}N^\alpha$, $\delta^{15}N^\beta$, $\delta^{15}N$ and $\delta^{18}O$-$N_2O$ should be given.

Response:

We have included a paragraph in the methods section that provides the values of standards used for analysis.

Manuscript Changes:

P4 L28- P5 L2 - "Our internal laboratory pure $N_2O$ tank standard (MSU Tank B) was isotopically characterized by analysis relative to the USGS51 and USGS52 reference materials (https://isotopes.usgs.gov/lab/referencematerials.html). Following the guidelines proposed by Coplen (2011), we report here the isotope values of the reference materials as well as our internal laboratory standard. The $\delta^{15}N$, $\delta^{18}O$, $\delta^{15}N^{\alpha}$, $\delta^{15}N^{\beta}$, and SP values of USGS51 and USGS52 are 1.32 ‰, 41.23 ‰, 0.48 ‰, 2.15 ‰, and -1.67 ‰ and 0.44 ‰, 40.64, 13.52, -12.64 ‰, and 26.15 ‰, respectively. The $\delta^{15}N$, $\delta^{18}O$, $\delta^{15}N^{\alpha}$, $\delta^{15}N^{\beta}$, and SP values of reference MSU Tank C are -0.9 ‰, 0.7 ‰, -2.6 ‰, 39.6 ‰, and 3.4 ‰, respectively. The $\delta^{15}N$, $\delta^{18}O$, $\delta^{15}N^{\alpha}$, $\delta^{15}N^{\beta}$, and SP values of the isotope standard MSU Tank B are -0.5 ‰, 11.13 ‰, -12.2 ‰, 40.8 ‰, and 23.3 ‰, respectively. All nitrogen isotope values are reported with respect to the international Air-$N_2$ standard, and all oxygen isotope values with respect to VSMOW."

P4 L31: How was f determined? From the amount of substrate provided minus the cumulative amount of $N_2O$ produced? Please add the respective information here.

Response:

We have included our mathematical approach to determining f.

Manuscript Change:

P5 L23-24 - "The fraction of substrate remaining was determined by dividing twice the amount of $N_2O$ produced by the total amount of nitrate added, and then subtracting this quantity from 1."

P6 section 3.1 and 3.2: Please provide results for $\eta^{15}N$ and $\eta^{18}O$ using the standard approach to calculate net isotope effects (e.g. $\delta^{15}N$-$N_2O$ – $\delta^{15}N$-$NO_3^-$).

Response:

Please see the response to Reviewer #1's general comments 1 and 4 for an explanation of the changes made to the manuscript.

Manuscript Change:

Please see the previous manuscript changes made as recommended in the response above.

P7 L5: One experiment with *P. aureofaciens* and succinate at 1 mM yields $N_2O$ with low SP similar to *P. chlororaphis*. How can this be explained or is it just an outlier?

Response:

We have changed the text to make the interpretation of our results reflect the reviewer's concerns.

Manuscript Change:

P7 L30-32 – The text has been changed to read "In four of the treatments, the average SP of *P. chlororaphis* denitrification was lower than that of *P. aureofaciens*. This resulted in a difference of 4.1‰ between the average SP of *P. chlororaphis* and *P. aureofaciens.*"

P7 L12 – 21: This section, the application of the Rayleigh model, should be preferably placed in the introductory or method section.

Response:

As recommended by Reviewer #2, we have moved the application of the Rayleigh model to the method section. Please see the text and below for the precise changes.

Manuscript Changes:

P5 L12-22 - "According to convention (Mariotti et al., 1981), the magnitude of the isotopic fractionation factor ($\alpha$) for a single unidirectional reaction is defined by the rate constants of the light ($k_1$) and heavy ($k_2$) isotopically substituted compounds:

$$\alpha = k_2/k_1. \tag{3}$$

Further, the isotopic enrichment factor, $\varepsilon$, is defined as

$$\varepsilon = (\alpha - 1) \times 1000, \tag{4}$$

and can be estimated from the slope of the linear relationship described by the Rayleigh model:

$$\delta^{15}N_p = \delta^{15}N_{so} - \varepsilon_{\underset{s}{p}}[(flnf)/(1-f)]; \tag{5}$$

where $\delta^{15}N_p$ is the isotope value of the accumulated product, $\delta^{15}N_{so}$ is the isotope value of the initial substrate, $\varepsilon$ is the fractionation factor, and f is the fraction of substrate remaining (Mariotti et al., 1981)."

P9 L14: The statement that SP depends on $\delta^{15}N^\alpha$ and $\delta^{15}N^\beta$ is trivial and could be omitted. The differences in SP between *Ps. aureofaciens* and *Ps. chlororaphis* are attributed to differences in the NO pool; which experiment could be used to check this hypothesis?

Response:

We have changed the text to reflect that these values are used to calculate SP rather than SP being related to them. As far as designing an experiment to determine if SP is affected by the NO pool, this is an inherently difficult problem, which arises from the difficulty in determining the concentration of NO available to the bacterium within the periplasmic space. We are currently beginning to examine NO reduction during enzymatic reduction both in vitro and in vivo.

Manuscript Changes:

See the response for Reviewer #1 comment 3; P10 L8 – P11 L11.

Page 8 (discussion): Temporal resolved data of $N_2O$ isotopic composition as used in this study could be preferably be collected using an online technique, e.g. laser spectroscopy – Please comment.

Response:

While online techniques such as off-axis integrated cavity output spectroscopy (OA-ICOS) and cavity ring down spectroscopy (CRDS) offer the capability to temporally resolve changes in $N_2O$ isotopomer values, the techniques have unique and challenges inherent to the technology to overcome. For instance, many obstacles remain that constrain the field, mesocosm, and *in vitro* application of OA-ICOS and CRDS including production and standardization of reference gases spanning a range of SP values and $N_2O$ concentrations, as well as the

standardization of methods for data analysis. Among other labs, our group is currently working to address challenges such as these as on-line measurements promise rapid continuous field-based and mesocosm data necessary for managing and monitoring $N_2O$ flux at local, ecosystem, and global scales. However, these topics have been reviewed and discusses in other peer-reviewed journals (Ostrom and Ostrom 2017). Therefore we feel that including such information detracts from the overall message we present in this manuscript.

Ostrom, N.E., Ostrom, P.H., 2017. Mining the isotopic complexity of nitrous oxide: a review of challenges and opportunities. Biogeochemistry, 132:3, 359-372.

Manuscript Changes:

Based upon the response above we do not feel that an in text response to the comment is necessary.

[revised manuscript text omitted]

Commented [JH7]: Reviewer #1: P5L14 Please see the comment I have written in the RefereeResponses document

Commented [JH8]: Reviewer #1: Specific comment P5L28

*aureofaciens* ranged from -77.5 ‰ to -18.4 ‰ and -106.2 ‰ to -11.4 ‰ for citrate and succinate, respectively, while values ranged from -119 ‰ to -9.2 ‰ and -82.1 ‰ to -5.1 ‰ for citrate and succinate, respectively, during denitrification by *P. chlororaphis* (Figure 2, Supplementary Table 1). Values of $\eta^{15}N$ greater than 0 were not observed. Such values would indicate that an inverse isotope effect is contributing fractionation, something that is not indicated by our data. Moreover, because *P. aureofaciens* and *P. chlororaphis* do not produce nitrous oxide reductase it is unlikely that an inverse isotope effect would be observed. Therefore, we have deliberately decided to cut off the density distributions at $\eta = 0$ ‰. Probability dDensity 
[revised manuscript text omitted]